# Anti-Inflammatory and Autophagy Activation Effects of 7-Methylsulfonylheptyl Isothiocyanate Could Suppress Skin Aging: In Vitro Evidence

**DOI:** 10.3390/antiox13111282

**Published:** 2024-10-23

**Authors:** Yeong Hee Cho, Jung Eun Park

**Affiliations:** 1Department of Biomedical Science, College of Natural Sciences and Public Health and Safety, Chosun University, Gwangju 61452, Republic of Korea; kdsinchi@naver.com; 2Department of Agricultural Biology, National Institute of Agricultural Sciences, RDA, Wanju-gun 55365, Republic of Korea; 3BK21-Four Educational Research Group for Age-Associated Disorder Control Technology, Chosun University, Gwangju 61452, Republic of Korea

**Keywords:** skin aging, 7-MSI, autophagy, inflammation, NLRP3

## Abstract

Skin inflammation, characterized by redness, swelling, and discomfort, is exacerbated by oxidative stress, where compounds like 7-methylsulfonylheptyl isothiocyanate (7-MSI) from cruciferous plants exhibit promising antioxidant and anti-inflammatory properties, though their effects on skin aging and underlying mechanisms involving the NLRP3 inflammasome and autophagy are not fully elucidated. NLRP3 is a crucial inflammasome involved in regulating inflammatory responses, and our study addresses its activation and associated physiological effects. Using biochemical assays such as ELISA, RT-qPCR, Western blotting, confocal microscopy, and RNA interference, we evaluated 7-MSI’s impact on cytokine production, protein expression, and genetic regulation in Raw 264.7 and RAW-ASC cells. 7-MSI significantly reduced TNF-α, IL-1β, IL-6, COX-2, and PGE transcription levels in LPS-stimulated Raw 264.7 cells, indicating potent anti-inflammatory effects. It also inhibited NF-κB signaling and NLRP3 inflammasome activity, demonstrating its role in preventing the nuclear translocation of NF-κB and reducing caspase-1 and IL-1β production. In terms of autophagy, 7-MSI enhanced the expression of Beclin-1, LC3, and Atg12 while reducing phospho-mTOR levels, suggesting an activation of autophagy. Moreover, it effectively decreased ROS production induced by LPS. The interaction between autophagy and inflammasome regulation was further confirmed through experiments showing that interference with autophagy-related genes altered the effects of 7-MSI on cytokine production. Collectively, this study demonstrates that 7-MSI promotes autophagy, including ROS removal, and to suppress inflammation, we suggest the potential use of 7-MSI as a skin care and disease treatment agent.

## 1. Introduction

Skin inflammation is a prevalent dermatological challenge that often presents with redness, swelling, and discomfort [1,2,3]. This condition not only affects the skin’s appearance but also has profound implications on its function and aging [1]. The aging process in skin is accelerated by various factors, including increased oxidative stress and inflammatory responses which lead to the breakdown of cellular structures and function [1,4,5]. Autophagy plays a crucial role in maintaining cellular homeostasis by degrading and recycling cellular debris and toxic substances, including reactive oxygen species (ROS) [2,6,7]. The removal of ROS is particularly vital as they are a major contributor to cellular aging and inflammation [8,9,10]. Moreover, autophagy is intricately linked to the regulation of inflammatory processes through these mechanisms that are yet to be fully elucidated.

As the skin ages, it undergoes significant immunological changes, characterized by a pro-inflammatory state driven by factors such as NF-κB and the inflammasome [11]. The activation of NF-κB triggers the release of pro-inflammatory cytokines and enzymes, which contribute to chronic inflammation in aging skin [12,13,14]. Additionally, the dysregulated activity of the inflammasome, particularly NLRP3, exacerbates inflammation and may lead to inflammatory skin disorders [15,16]. The NLRP3 inflammasome is a key component of the immune response, playing a central role in the innate immune defense against pathogens and cellular damage [17,18,19]. However, its activation is also linked to various inflammatory diseases and age-related pathological conditions [16]. Therefore, controlling NLRP3 inflammasome activity could be crucial for mitigating inflammatory responses in the skin.

Isothiocyanate, a compound found in cruciferous vegetables, has garnered attention due to its antioxidant, anti-inflammatory, and anticancer properties (Figure 1) [20,21,22]. Despite these benefits, the effects of 7-methylsulfonylheptyl isothiocyanate (7-MSI) on skin aging and its underlying mechanisms have not been thoroughly explored. This study investigates the impact of 7-MSI on autophagy and inflammatory responses in skin cells, particularly through its interaction with the NLRP3 inflammasome.

In this study, we aim to investigate the biological activity of the natural antioxidant 7-MSI, focusing on its inhibitory effects on inflammatory responses, inflammasome formation, and autophagy activation in murine macrophage cell lines (Raw 264.7 and RAW-ASC cells). By elucidating the mechanisms of 7-MSI, we hope to uncover its therapeutic potential in targeting skin aging and related inflammatory conditions, offering new insights into its clinical applications in dermatology.

## 2. Materials and Methods

### 2.1. Cell Culture

Raw 264.7 macrophages were purchased from American Type Culture Collection (ATCC) (Manassas, VA, USA) and RAW-ASC cells were purchased from InvivoGen (San Diego, CA, USA). The cells were cultured in Dulbecco’s modified Eagle medium (DMEM) (Lonza, Switzerland) supplemented with 10% fetal bovine serum (FBS) (Atlas Biologicals, Fort Collins, CO, USA), 1% penicillin-streptomycin (Sigma-Aldrich, St. Louis, MO, USA), 50 μg/mL blasticidin, and 100 μg/mL normocin in a 5% CO_2_ incubator at 37 °C.

### 2.2. Cell Viability Assay

Cell viability was assessed using the CellTiter 96^®^ AQueous Non-Radioactive Cell Proliferation Assay (Promega, Madison, WI, USA). RAW 264.7 cells were seeded in 96-well plates at a density of 5 × 10^4^ cells per well and allowed to adhere overnight. Subsequently, the cells were exposed to varying concentrations of 7-MSI (0, 0.1, 0.5, 1, 3, and 5 μg/mL) and TGF-β (0, 0.00002, 0.0002, 0.002, 0.02, and 0.2 ng/mL) for a period of 24 h. Following the treatment, 20 μL of MTS reagent was added to each well. The plates were then incubated at 37 °C with 5% CO_2_ for 4 h. Optical density was subsequently measured at 490 nm using a microplate reader.

### 2.3. Western Blot Analysis

Raw 264.7 and RAW-ASC cells were seeded at a density of 2.0 × 10^5^ cells/well in 6-well plates. After overnight culture, the Raw 264.7 cells were treated with LPS (0.01 or 1 μg/mL; Sigma-Aldrich) with or without 7-MSI (1 or 2 μg/mL; LKT Laboratories, St. Paul, MN, USA) for 0.5, 1, and 3 h at 37 °C in 5% CO_2_ to induce an inflammatory response and inflammasome formation. The RAW-ASC cells were primed with LPS (0.01 μg/mL) for 3 h, and then treated with 7-MSI (2 μg/mL) and ATP (1 mM; InvivoGen) for an additional 3 h at 37 °C in 5% CO_2_. Protein extracts (50 or 100 μg) were obtained using ProNA CETi lysis buffer (TransLab, Daejeon, Republic of Korea), boiled at 100 °C for 3 min, and subjected to SDS-PAGE on 8%, 10%, 12%, or 15% gels. The proteins were then transferred onto polyvinylidene fluoride (PVDF) membranes (Bio-Rad, Hercules, CA, USA). The membranes were blocked with 5% skim milk in TBS-T (250 mM Tris-HCl, pH 8.0, 1.5 mM NaCl, and 0.1% Tween 20) at room temperature for 2 h. Subsequently, the membranes were incubated overnight at 4 °C with primary antibodies (1:500 or 1:1000 in blocking buffer) against NF-κB-p65 (Santa Cruz Biotechnology, Dallas, TX, USA), GAPDH, Beclin-1, Atg12-5, mTOR, phospho-mTOR, IκB-α, phospho-IκB-α, ASC, NLRP3, and caspase-1 (Cell Signaling Technology, Danvers, MA, USA). After six washes with TBS-T buffer, the membranes were incubated with horseradish peroxidase-conjugated anti-rabbit or mouse IgG antibodies (1:4000 in blocking buffer; Cell Signaling Technology, Danvers, MA, USA) at room temperature for 2 h. Following five additional washes with TBS-T buffer, protein expression levels were detected using EZ-Western Lumi Plus and Lumi Femto (DAEILLAB SERVICE, Seoul, Republic of Korea), and visualized on X-ray film (Fuji Film, Tokyo, Japan). Band intensities were quantified using ImageJ software V1.8.0 (National Institutes of Health, Bethesda, MD, USA).

### 2.4. Enzyme-Linked Immunosorbent Assay (ELISA)

To examine the anti-inflammatory properties of 7-MSI, Raw 264.7 and RAW-ASC cells were plated in 24-well plates at a density of 0.25 × 10^5^ cells/well and incubated overnight. Following incubation, the cells were treated with varying concentrations of 7-MSI (0–4 µg/mL) in the presence of LPS (1 µg/mL) for 1 h at 37 °C in a 5% CO_2_ atmosphere. To quantify TNF-α levels in the culture supernatants, an ELISA kit (R&D Systems, Minneapolis, MN, USA) was used, following the manufacturer’s instructions. Absorbance was measured at 450 nm with a microplate reader (Molecular Devices, San Jose, CA, USA). Additionally, RAW-ASC cells were seeded in 6-well plates at a density of 2 × 10^5^ cells/well and allowed to incubate overnight. These cells were then primed with LPS (0.01 μg/mL) for 3 h, followed by treatment with 7-MSI (2 μg/mL) and ATP (1 mM) for another 3 h at 37 °C in 5% CO_2_. The levels of IL-1β in the culture supernatants were assessed using an ELISA kit (R&D Systems) according to the manufacturer’s protocol, with absorbance readings taken at 450 nm using a microplate reader (Molecular Devices, San Jose, CA, USA).

### 2.5. Measurement of ROS Production

Raw264.7 cells were cultured at a density of 1 × 10^4^ cells/well in 6-well plates for 24 h, and then treated with LPS (1 μg/mL; Sigma-Aldrich) and either with or without 7-MSI (2 μg/mL; LKT Laboratories, St. Paul, MN, USA) or NAC (0.82 mg/mL) for 1 h at 37 °C in a 5% CO_2_ using serum-free medium. The cells were stained with 5 μM of CM-H_2_DCF-DA (Invitrogen, Waltham, MA, USA) and incubated for 30 min at 37 °C in a 5% CO_2_ incubator, and the fluorescence intensities were measured at excitation and emission wavelengths of 488 and 530 nm, respectively. After incubation, the cells were collected using trypsin-EDTA, washed with PBS, and resuspended in 400 μL of PBS. The FACS analysis was conducted on a BD FACS Calibur (Becton Dickinson, Fanklin Lakes, NJ, USA) and data were analyzed using the BD CellQuest Pro software v4.0.2.

### 2.6. Reverse Transcription-Quantitative Polymerase Chain Reaction (RT-qPCR)

To isolate total RNA, RAW 264.7 cells were lysed, and the RNeasy Plus Mini Kit (QIAGEN, Hilden, Germany) was used, following the manufacturer’s protocol. For cDNA synthesis, 500 µg of total RNA was mixed with 33 pmol of oligo (dT)^18^ primers, 10 mM dNTPs, and M-MLV reverse transcriptase (Bioneer, Daejeon, Korea), and the reaction was carried out at 25 °C for 5 min, 42 °C for 1 h, and 70 °C for 15 min. For the qPCR amplification of the target genes, synthesized cDNA (1–100 ng) served as the template, utilizing Top DNA polymerase (Bioneer, Daejeon, Republic of Korea). The cycling conditions were as follows: denaturation at 94 °C for 30 s; annealing at 48 °C, 55 °C, or 60 °C for 30 s; and extension at 72 °C for 30 s, repeated for 30 cycles. The primer sequences used for qPCR are listed in Appendix A.

### 2.7. Immunostaining and Confocal Microscopic Analysis

Raw 264.7 or RAW-ASC cells (1.0 × 10^5^ cells) were seeded on poly-L-lysine (0.01% solution)-coated glass coverslips in 12-well plates and cultured at 37 °C in 5% CO_2_. After 24 h, Raw 264.7 cells were treated with LPS (1 μg/mL) and 7-MSI (1 μg/mL) for 1 h, while RAW-ASC cells were primed with LPS (0.01 μg/mL) for 3 h followed by treatment with 7-MSI (2 μg/mL) and ATP (1 mM) for an additional 3 h. The cells were then washed with 0.01 M pH 7.4 phosphate-buffered saline (PBS) and fixed with 3.7% formaldehyde in PBS for 10 min at room temperature. After fixation, the cells were permeabilized with 0.1% Triton X-100 for 10 min and washed three times with PBS. Blocking was performed with 1% bovine serum albumin (BSA) in PBS at room temperature for 20 min. The cells were then incubated with anti-MAP LC3-II, anti-NF-κB p65, anti-ASC, or anti-Caspase-1 antibodies (1:50 in PBS; Santa Cruz Biotechnology, Dallas, TX, USA) for 1 h at room temperature. Following primary antibody incubation, the cells were washed three times with PBS and incubated with Alexa Fluor 488-conjugated goat anti-mouse IgG (1:200 in PBS; Invitrogen, Waltham, MA, USA) or goat anti-rabbit IgG-TRITC (1:200 in PBS; Santa Cruz Biotechnology, Dallas, TX, USA) secondary antibodies for 1 h at room temperature. After being washed three times with PBS, the cells were stained with 4′,6-diamidino-2-phenylindole (DAPI). The stained cells were than observed under a Zeiss LSM 510 confocal microscope (Le Pecq, France).

### 2.8. Caspase-1 Activity Assay

RAW-ASC cells were seeded at a density of 2 × 10^5^ cells/well in 6-well plates and cultured overnight. The cells were primed with LPS (0.01 μg/mL) for 3 h, followed by treatment with 7-MSI (2 μg/mL) and ATP (1 mM) for an additional 3 h at 37 °C in 5% CO_2_. Caspase-1 activity was assessed using a fluorometric caspase-1 assay kit (Abcam, Cambridge, UK) according to the manufacturer’s instructions. Briefly, the treated cells were washed with cold PBS and resuspended in 50 μL of cell lysis buffer, followed by incubation on ice for 10 min. The cell lysates were then centrifuged at 16,000× *g* for 4 min at 4 °C, and the supernatants were collected. Equal volumes (50 μL) of the samples and 2× reaction buffer (containing 10 mM dithiothreitol) were mixed, and 50 μM of YVAD-AFC substrate was added. After incubating at 37 °C for 1 h, fluorescence intensity was measured using a microplate reader (Molecular Devices) at excitation and emission wavelengths of 400 nm and 505 nm, respectively.

### 2.9. RNA Interference

RAW-ASC cells were transfected with 400 pmol of Atg5 and Beclin-1 small interfering RNA (siRNA; Bioneer, Daejeon, Korea) using a siRNA transfection reagent. The transfection was performed in siRNA transfection medium (serum-free) at 37 °C in 5% CO_2_ for 24 h. As a control, non-transfected cells were included and handled in the same manner. Following transfection, both control and transfected cells were subjected to the treatments and analyses as previously described [23].

### 2.10. Statistical Analysis

All statistical data and graphs were analyzed using SigmaPlot v10.0 (Systat Software Inc., San Jose, CA, USA). Data are expressed as mean ± standard error of mean, and statistical significance was determined by Student’s *t*-test using SigmaPlot software v10.0; *p* < 0.05 was considered to indicate statistical significance.

## 3. Results

### 3.1. Effect of 7-MSI on Cell Viability in RAW 264.7

The potential influence of 7-MSI on cell viability was investigated using an MTS assay in RAW 264.7 cells treated with varying doses of 7-MSI (ranging from 0 to 5 µg/mL) over a 24 h period. According to the results depicted in Appendix A, 7-MSI exhibited no cytotoxic effects at concentrations up to 5 µg/mL. These findings indicate that 7-MSI does not adversely affect the viability of RAW 264.7 cells.

### 3.2. Anti-Inflammatory Effects of 7-MSI in Raw 264.7 Cells

ELISA analysis demonstrated a significant increase in TNF-α production from 55.47 pg/mL to 1106 pg/mL in Raw 264.7 cells treated with 1 µg/mL of LPS for 1 h. However, treatment with 7-MSI (1 μg/mL) reduced TNF-α levels to 752 pg/mL compared to the LPS-stimulated cells (Figure 2A). RT-qPCR further revealed that 7-MSI (0, 0.5, 1, and 2 μg/mL) significantly decreased the transcription levels of IL-1β, IL-6, COX-2, and PGEs induced by LPS (1 μg/mL) by approximately 562%, 126%, 209%, and 56%, respectively, compared to the LPS only control group (Figure 2B,C). These results suggest that 7-MSI inhibits the LPS-induced inflammatory response by suppressing pro-inflammatory cytokine production.

### 3.3. 7-MSI Suppresses NF-κB Activation in Raw 264.7 Cells

Western blotting demonstrated that 7-MSI (1 µg/mL) treatment reduced the degradation of IκBα and significantly decreased phospho-IκBα production by approximately 46% in LPS-stimulated Raw 264.7 cells (Figure 2D–F). Confocal microscopy using fluorescein-labeled anti-NF-κB antibody confirmed that 7-MSI inhibited the nuclear translocation of NF-κB protein induced by LPS, retaining NF-κB in the cytoplasm (Figure 2G). These findings suggest that 7-MSI suppresses NF-κB signaling by inhibiting IκBα phosphorylation and subsequent degradation.

### 3.4. 7-MSI Suppresses NLRP3 Inflammasome Activity in RAW-ASC Cells

RT-qPCR analysis showed that 7-MSI treatment significantly reduced the transcription levels of NLRP3, caspase-1, and IL-1β by approximately 72%, 65%, and 10%, respectively, compared to the non-treated LPS-primed plus ATP group (Figure 3A,B). The expression levels of NLRP3, ASC, and pro-caspase-1 decreased by 80%, 34%, and 86%, respectively, in RAW-ASC cells primed with ATP (1 mM) and LPS (0.01 μg/mL) following treatment with 2 µg/mL of 7-MSI compared to those of the non-7-MSI-treated cells (Figure 3C,D). Immunostaining and confocal microscopy further confirmed that 7-MSI suppressed inflammasome formation in RAW-ASC cells primed with LPS and ATP (Figure 3E). Additionally, caspase-1 activity decreased by 37.35%, and IL-1β production was reduced by approximately 91.83% in 7-MSI-treated cells compared to controls (Figure 4A,B). These results indicate that 7-MSI inhibits IL-1β release and suppresses NLRP3 inflammasome activation.

### 3.5. 7-MSI Activates Cellular Autophagy in Murine Macrophages

Treatment with 7-MSI (1 µg/mL) in the presence of LPS (1 μg/mL) increased the expression levels of autophagy-associated proteins Beclin-1, LC3, and Atg12 by approximately 377%, 27%, and 77%, respectively, while decreasing phospho-mTOR levels by approximately 34% compared to LPS stimulation alone (Figure 5A,B). Immunostaining with anti-LC3II antibody confirmed the formation of autophagosomes in 7-MSI-treated cells (Figure 5C). As shown in Figure 5D, LPS increased ROS production up to 130.67%, compared to that of non-irradiated control. However, the levels of ROS production could be decreased by 7-MSI and NAC treatments to approximately 29.74% and 33.31%, respectively, compared to that of the LPS only group. These results suggest that 7-MSI reduced LPS-induced ROS production by approximately 29.74% compared to the LPS only group (Figure 5D), suggesting enhanced autophagic activity (Figure 5A–C) and reduced oxidative stress (Figure 5D). 

### 3.6. Interaction of Autophagy and Inflammasome Regulation by 7-MSI

Finally, we investigated the relationship between the activation of the autophagy system and the inhibition of inflammasome formation by 7-MSI using RNA interference experiments. The ELISA results of RAW-ASC cells treated with LPS (0.01 μg/mL) and 7-MSI (2 μg/mL) showed a 26% decrease in the production of TNF-*α*. However, in cells transfected with Beclin-1 siRNA, the production level increased by the same amount as in the group not treated with 7-MSI (Figure 6A). In addition, the analysis of IL-1β secretion, produced by NRLP3 activation, through ELISA revealed that the secretion of IL-1β decreased by 92.02% due to 7-MSI treatment. However, in cells transfected with Beclin-1 siRNA, the secretion increased almost equivalently to the group not treated with 7-MSI (Figure 6B).

## 4. Discussion

The findings from this study collectively demonstrate the antioxidant and anti-inflammatory properties of 7-MSI, shedding light on its mechanisms of action and potential therapeutic applications. This study highlights several key aspects of the biological activity of 7-MSI, providing valuable insights into its potential as a therapeutic agent for inflammatory and dermatological conditions. Specifically, this discussion focuses on the inhibition of pro-inflammatory cytokines, the suppression of NF-κB activation, the inhibition of NLRP3 inflammasome activity, the induction of autophagy leading to a reduction in ROS, and the interplay between autophagy and inflammasome regulation.

7-MSI effectively inhibits the production of pro-inflammatory cytokines TNF-α (Figure 2A) and the transcription level of IL-1β, IL-6, COX-2, and PGEs in Raw 264.7 and RAW-ASC cells (Figure 2B,C). These cytokines are crucial mediators of the inflammatory response [24,25,26], and their reduction signifies the potential of 7-MSI in mitigating inflammatory processes. This inhibition is significant, as it highlights the ability of 7-MSI to interfere with critical pathways involved in inflammation. NF-κB, a key transcription factor regulating inflammation-related genes [13,27], is suppressed by 7-MSI through the inhibition of IκBα degradation and reduction in phospho-IκBα production (Figure 2D–G). This suppression indicates that 7-MSI plays a role in modulating NF-κB signaling pathways, which are pivotal in skin inflammation. This inhibition limits the nuclear translocation of NF-κB and reduces the expression of pro-inflammatory mediators, thereby alleviating inflammatory processes exacerbated by oxidative stress. The NLRP3 inflammasome is a crucial mediator of inflammation and oxidative stress-related disorders [17,19,20,28]. Our findings demonstrate that 7-MSI inhibits NLRP3 inflammasome activation (Figure 3), thereby attenuating IL-1β production and caspase-1 activity (Figure 4). These observations highlight the potential of 7-MSI to mitigate inflammatory responses associated with NLRP3 inflammasome activation, thereby ameliorating oxidative stress-induced tissue damage.

Autophagy is a crucial cellular process involved in maintaining redox homeostasis [29,30] and cellular integrity [2,6,29,30,31]. Our study demonstrates that 7-MSI enhances autophagic activity in Raw 264.7 cells by upregulating the expression of Beclin-1, LC3, and Atg12 proteins, while downregulating phospho-mTOR levels (Figure 5A–C). This induction of autophagy is associated with decreased ROS production (Figure 5D) and enhanced cellular antioxidant defenses. These findings suggest that 7-MSI promotes cellular resilience against oxidative stress through the activation of autophagy pathways. Furthermore, our RNA interference experiments targeting Beclin-1 underscore the interplay between autophagy and inflammasome regulation by 7-MSI (Figure 6), altering its effects on cytokine production and highlighting the interconnected nature of these pathways in mediating its anti-inflammatory effects. These findings provide insights into the mechanisms underlying 7-MSI’s therapeutic potential in managing skin inflammation and aging-related conditions.

The comprehensive characterization of 7-MSI’s anti-inflammatory and antioxidant properties underscores its promise as a therapeutic agent for dermatological conditions characterized by inflammation and oxidative stress. Furthermore, the ability of 7-MSI to modulate key pathways involved in inflammation and cellular stress suggests its potential applicability in treating skin aging and inflammatory skin disorders.

In summary, the findings presented in this study highlight the multifaceted biological activity of 7-MSI as a natural product (Figure 7). It effectively suppresses pro-inflammatory cytokine production, inhibits NF-κB activation and NLRP3 inflammasome activity, and induces cellular autophagy. These actions demonstrate 7-MSI’s potential in addressing skin inflammation and aging-related conditions. This study also reveals the interplay between autophagy induction and inflammasome regulation, offering new insights into the mechanisms of action of 7-MSI. Future research should focus on further elucidating the precise molecular mechanisms of 7-MSI and evaluating its efficacy and safety in vivo. Additionally, clinical studies exploring its applicability in dermatological settings are essential to fully realize the therapeutic potential of 7-MSI in treating skin inflammation and aging-related disorders. The comprehensive characterization of 7-MSI’s anti-inflammatory properties underscores its promise as a novel therapeutic strategy for dermatological conditions characterized by inflammation.

## Figures and Tables

**Figure 1 antioxidants-13-01282-f001:**
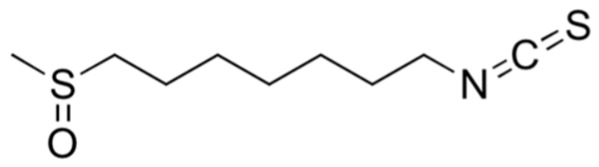
Structure of 7-MSI.

**Figure 2 antioxidants-13-01282-f002:**
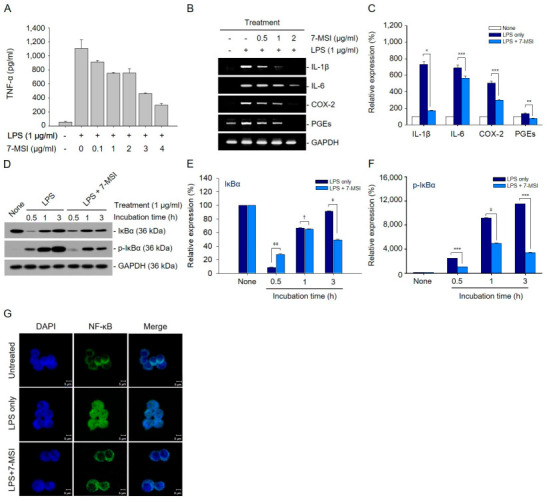
Anti-inflammatory effect of 7-MSI. (**A**) Raw 264.7 cells were exposed to varying concentrations of 7-MSI (0-4 μg/mL) in the presence of LPS (1 μg/mL) for 1 h. TNF-α levels in the culture supernatants were quantified using a specific ELISA kit for TNF-α and were compared to a standard curve for accuracy. (**B**) Raw 264.7 cells were exposed to 7-MSI at concentrations of 0.5, 1, and 2 μg/mL, alongside LPS at 1 μg/mL, for a duration of 3 h. Total RNA was extracted, and RT-qPCR was performed using primers that specifically target IL-1β, IL-6, COX-2, and PGEs. (**C**) The histograms display the relative expression rates of IL-1β, IL-6, COX-2, and PGEs, calculated as the ratio of signal intensity to GAPDH. (**D**) RAW 264.7 cells were exposed to 7-MSI (1 μg/mL) alongside LPS (1 μg/mL) for durations of 0.5, 1, and 3 h. Protein expression levels of IκBα, p-IκBα, and GAPDH were assessed through Western blotting. (**E**,**F**) The histograms illustrate the relative expression rates of IκBα and p-IκBα, normalized to the signal intensity of GAPDH. The data represent mean ± SD from triplicate Western blot experiments. Statistical significance was analyzed using Student’s *t*-test with the following significance levels: * *p* < 0.00001; ** *p* < 0.00005; *** *p* < 0.0001; ^ǂ^ *p* < 0.0005; ^ǂǂ^ *p* < 0.001; ^†^ *p* < 0.05, compared to the “LPS only” group. (**G**) Cells were treated with LPS (1 μg/mL) alone for 1 h or in combination with 7-MSI (1 μg/mL) for 1 h, and then stained with a fluorescein-conjugated antibody targeting NF-κB p65. The green fluorescence was subsequently observed using a confocal microscope.

**Figure 3 antioxidants-13-01282-f003:**
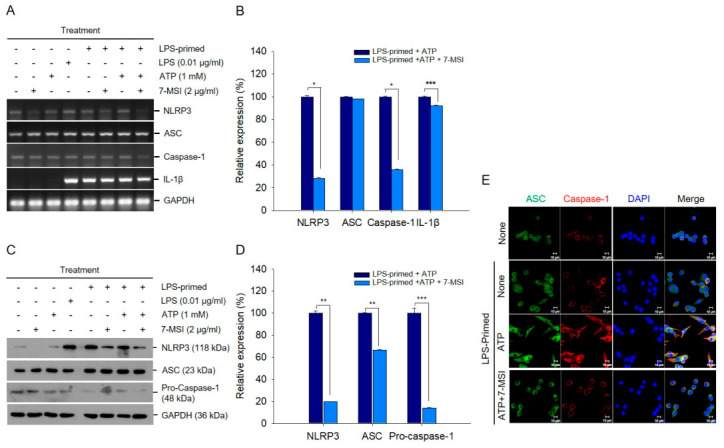
7-MSI suppresses NLRP3 inflammasome activity in RAW-ASC. (**A**) RAW-ASC cells were first primed with LPS (0.01 μg/mL) for 3 h, followed by treatment with 7-MSI (2 μg/mL) and ATP (1 mM) for an additional 3 h. Western blotting was performed to assess the expression levels of ASC, NLRP3, pro-caspase-1, and GAPDH. (**B**) The histograms display the relative expression levels of ASC, NLRP3, and pro-caspase-1 proteins, with each value calculated as the signal intensity ratio to GAPDH. (**C**) The expression levels of NLRP3, ASC, caspase-1, IL-1β, and GAPDH were measured using RT-qPCR. The cells were primed with LPS (0.01 μg/mL) for 3 h and subsequently treated with 7-MSI (2 μg/mL) and ATP (1 mM) for another 3 h. (**D**) The histograms depicting the relative expression rates from the RT-qPCR analysis. Each value was determined by comparing the signal intensity ratio to that of GAPDH. The symbols “+” and “-” indicate the presence or absence of the respective treatments. The histograms show the means ± S.D. from two independent experiments. Statistical significance was evaluated using Student’s *t*-test with the following significance levels: * *p* < 0.0001; ** *p* < 0.0005; *** *p* < 0.005, compared to the “LPS-primed + ATP” group. (**E**) RAW-ASC cells treated with LPS (0.01 μg/mL) for 3 h and then with 7-MSI (2 μg/mL) and ATP (1 mM) for an additional 3 h were stained with fluorescein-labeled anti-ASC and anti-Caspase-1 antibodies. The green and red fluorescences emitted were observed using a confocal microscope.

**Figure 4 antioxidants-13-01282-f004:**
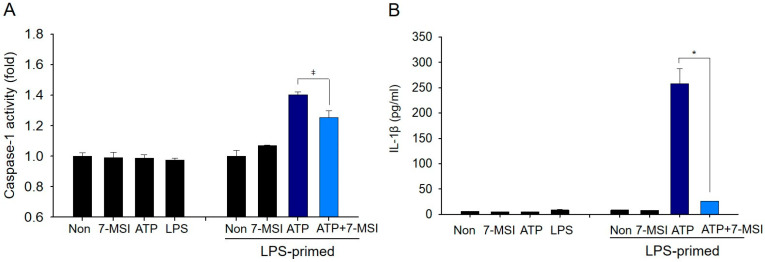
Inhibitory effects of 7-MSI on caspase-1 activity and IL-1β production in RAW-ASC cells. (**A**) Caspase-1 activity measured in RAW-ASC cells. The cells were first primed with LPS (0.01 μg/mL) for 3 h, followed by co-treatment with 7-MSI (2 μg/mL) and ATP (1 mM) for an additional 3 h. Caspase-1 activity was assessed using a specific assay kit. (**B**) IL-1β concentrations in the culture supernatants. Following the same treatment protocol as in Panel A, IL-1β levels were quantified using an ELISA kit specific for IL-1β. The results represent the means ± standard deviation (SD) of duplicate determinations from three independent experiments. Statistical significance was assessed using a Student’s *t*-test, with significance levels indicated by the following symbols: * for *p* < 0.005 and ǂ for *p* < 0.5, both compared to the “LPS-primed + ATP” group. The data indicate that 7-MSI potentially inhibits both caspase-1 activity and IL-1β production under the conditions tested.

**Figure 5 antioxidants-13-01282-f005:**
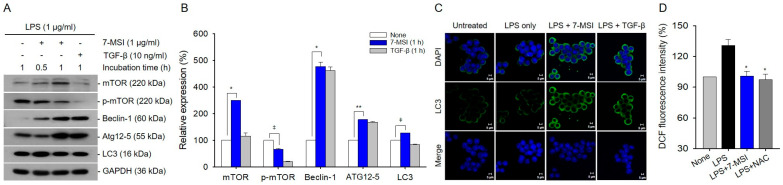
7-MSI induces activation autophagy in Raw 264.7 cells. (**A**) Raw 264.7 cells were exposed to 7-MSI (1 μg/mL) along with LPS (1 μg/mL) for either 30 min or 1 h. Western blotting was utilized to assess the expression levels of mTOR, phosphorylated mTOR (p-mTOR), Beclin-1, Atg12, LC3, and GAPDH. TGF-β (10 ng/mL) was used as a positive control for the experiment. (**B**) The histograms show the relative expression levels of the proteins from the 1 h treatment group in panel A. Each value was calculated as the ratio of signal intensity to that of GAPDH, ensuring accurate normalization and comparison. The symbols “+” and “−” denote the inclusion and omission of the respective treatments. The data represent the means ± S.D. values of the results of two separate experiments. The statistical significance of the data was determined using a Student’s *t*-test, with significance levels indicated as follows: * *p* < 0.0001; ** *p* < 0.0005; ǂ *p* < 0.005, in comparison to the “none” group, which was treated with LPS only. (**C**) Cells treated with LPS (1 μg/mL) for 1 h or co-treated with LPS (1 μg/mL) and 7-MSI (1 μg/mL) for 1 h were stained with a fluorescein-labeled antibody against LC3-II. Green fluorescence, indicating the presence of LC3-II, was visualized using a confocal microscope, allowing for the observation of autophagy-related processes in the cells. (**D**) To measure ROS levels, the cells were then further incubated with DCFH-DA (20 μM) for 30 min at 37 °C and the fluorescence intensities were measured using flow cytometry. * *p* < 0.0005, in comparison to the “LPS” group.

**Figure 6 antioxidants-13-01282-f006:**
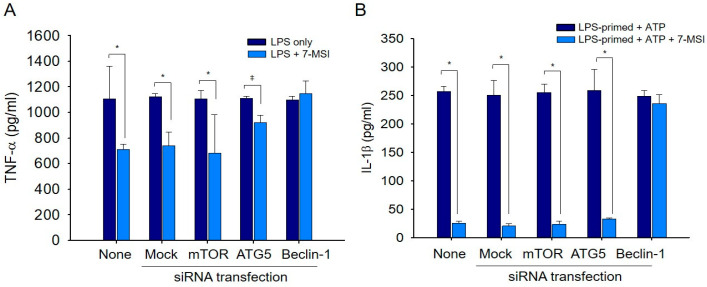
Effect of ATG5 and Beclin-1 knockdowns on inflammatory response in RAW-ASC cells. (**A**) Quantification of TNF-α levels in culture supernatants of RAW-ASC cells transfected with siRNAs targeting none, mock, Beclin-1, mTOR, and ATG-5. Post-transfection, cells were primed with LPS (0.01 μg/mL) for 3 h and subsequently treated with 7-MSI (2 μg/mL) and ATP (1 mM) for an additional 3 h. TNF-α levels were measured using a specific ELISA kit. (**B**) Measurement of IL-1β levels in the culture supernatants following the same treatment as described for Panel A. IL-1β was quantified using an IL-1β-specific ELISA kit. Statistical analyses were performed using a Student’s *t*-test, with the significance levels denoted as * for *p* < 0.005 and ǂ for *p* < 0.05, both compared to the “ATP” group (LPS-primed group). The “None” label refers to the non-transfected control group. This figure demonstrates how the knockdown of specific autophagy-related genes influences the production of inflammatory cytokines in response to LPS and ATP treatments, adjusted with 7-MSI.

**Figure 7 antioxidants-13-01282-f007:**
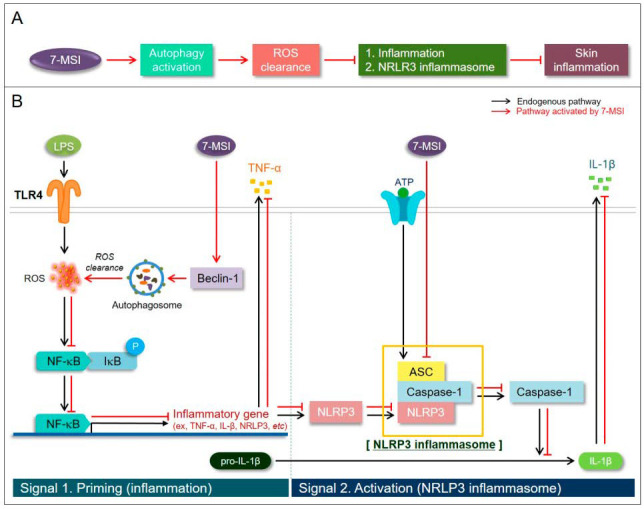
Schematic summary of 7-MSI on anti-inflammation and autophagy activation. (**A**) summarizes the effect of 7-MSI on autophagy and reactive oxygen species (ROS) clearance. 7-MSI activates autophagy and reduces ROS, which in turn can lead to general inflammation or specifically trigger the NLRP3 inflammasome, contributing to skin inflammation. (**B**) illustrates the molecular pathways influenced by LPS through TLR4, highlighting how 7-MSI intersects with these pathways. In the priming stage, the expression of NLRP3 and pro-IL-1β is increased, while the activation stage involves the formation of the ASC and caspase-1 complex, leading to the production of active caspase-1 and IL-1β. 7-MSI modulates these pathways at various points, potentially reducing the inflammatory response by inhibiting the activation of key inflammatory mediators.

## Data Availability

The original contributions presented in the study are included in the article/Appendix A, further inquiries can be directed to the corresponding author.

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
