# Peer review of "Anti-Inflammatory and Autophagy Activation Effects of 7-Methylsulfonylheptyl Isothiocyanate Could Suppress Skin Aging: In Vitro Evidence"

_antioxidants, 2024, doi:10.3390/antiox13111282_

Round 1

Reviewer 1 Report

The authors studied the effects of the plant-derived substance 7-methylsulfonylheptyl isothiocyanate (7-MSI) on the murine macrophage cell lines Raw 264.7 and RAW-ASC. The results suggest anti-inflammatory activities. The authors conclude in the title that “Anti-inflammatory and autophagy activation effects of 7-methylsulfonylheptyl isothiocyanate could suppress skin aging.”

The title is not supported by the data. Macrophage cell lines are not a model for the skin. The study was not designed to obtain insights into aging. The title and the interpretation of the results must be changed.

Several statements in the text are not scientifically precise and conclusive. Example: Abstract, first sentence “Skin inflammation, … is exacerbated by … inflammation ...” Much of the text needs to be rephrased.

The potential toxicity of 7-MSI at the chosen concentrations needs to be determined.

Figure 3B: The vertical scale does not allow to determine the values of the bar chart.

The activation of autophagy should be tested with a valid assay, for example change of LC3-I : LC3-II ratio. Figure 4 is not conclusive.

The change of beclin in response to 7-MSI is interesting. Quantification of protein (including statistics) would improve this data. How many times was the experiment repeated?

More references should be provided to show the state of knowledge.

Author Response

Major comments

The authors studied the effects of the plant-derived substance 7-methylsulfonylheptyl isothiocyanate (7-MSI) on the murine macrophage cell lines Raw 264.7 and RAW-ASC. The results suggest anti-inflammatory activities. The authors conclude in the title that “Anti-inflammatory and autophagy activation effects of 7-methylsulfonylheptyl isothiocyanate could suppress skin aging.”

Comments 1: The title is not supported by the data. Macrophage cell lines are not a model for the skin. The study was not designed to obtain insights into aging. The title and the interpretation of the results must be changed.

Response 1: According to Reviewers (Reviewer 1 & 2) comment, we have revised the title as follows: "Anti-inflammatory and autophagy activation effects of 7-methylsulfonylheptyl isothiocyanate could suppress skin aging: In vitro evidence.”

Detail comments

Comments 1: Several statements in the text are not scientifically precise and conclusive. Example: Abstract, first sentence “Skin inflammation, … is exacerbated by … inflammation ...” Much of the text needs to be rephrased.

Response 1: Thank you for your valuable comment. We have modified the sentence “Skin inflammation, characterized by redness, swelling, and discomfort, is exacerbated by oxidative stress, where compounds like 7-methylsulfonylheptyl isothiocyanate (7-MSI) from cruciferous plants exhibit promising antioxidant and anti-inflammatory properties, though their effects on skin aging and underlying mechanisms involving the NLRP3 inflammasome and autophagy are not fully elucidated.” and marks in red.

Comments 2: The potential toxicity of 7-MSI at the chosen concentrations needs to be determined.

Response 2: Thank you for your valuable comment. We have added new figure as a Supplementary Figure 1.

Comments 3: Figure 3B: The vertical scale does not allow to determine the values of the bar chart.

Response 3: In response to the reviewer's comment, we have adjusted the vertical scale of the bar chart to clearly display the values, thereby enhancing visibility and ensuring that the values are easily identifiable. Revised as Figure 4B.

Comments 4: The activation of autophagy should be tested with a valid assay, for example change of LC3-I: LC3-II ratio. Figure 4 is not conclusive.

Response 4: Thank you for your insightful comment regarding our study's test for autophagy activation. We agree that the LC3-I to LC3-II ratio is a definitive indicator of autophagy. However, in cases involving LC3 enrichment, the LC3-I and LC3-II bands are not always completely resolved, although an increase in LC3 is considered directly related to the activation of autophagy. Unfortunately, due to time constraints, we are unable to perform a detailed analysis of LC3 enrichment at this stage. We greatly value your comment and plan to incorporate this into subsequent research to further our understanding of the autophagic process. We ask for your understanding in this matter.

Comments 5: The change of beclin in response to 7-MSI is interesting. Quantification of protein (including statistics) would improve this data. How many times was the experiment repeated?

Response 5: Thank you for your comments, and we appreciate your emphasis on the significance of the change in Beclin levels in response to 7-MSI treatment. We agree that the variations in Beclin expression are particularly intriguing due to the differences between cell lines. We recognize the importance of robust quantification of protein changes and providing statistical validation of the data. To ensure reproducibility, the experiment was conducted twice.

Comments 6: More references should be provided to show the state of knowledge.

Response 6: Thank you for your suggestion. We have incorporated additional references into the revised manuscript as per your recommendation. We appreciate your guidance in enhancing the scholarly depth of our work.

Reviewer 2 Report

The manuscript by Cho and Park on 7MSI for anti-inflammatory and autophagy effects on skin represents an exploratory in vitro investigation.  Several points should be addressed.

1. Title, this is an in vitro study, please change the title to:

Anti-inflammatory and autophagy activation effects of 7-methylsulfonylheptyl isothiocyanate could suppress skin aging: In vitro evidence

2. Abstract – define NLRP3

3.  The description of 7MSI is not clear, please generate a figure with the chemical structure of this compound

a. References 21 and 22 about isothiocyanates is lacking, please include Navarro et al, Food & Function, 2011 as a reference

b. There is no description of the abundance 7MSI in cruciferous vegetables

c.  If 7MSI can penetrate the skin layers

d. How stable 7MSI is or what is its solubility, etc.

4. How do the authors justify the high concentrations of 7MSI used in this in vitro study (micrograms/ml), while on the other hand propose that this compound may be a potential active ingredient for cosmetics?  Please include in the discussion the limitations of this in vitro study.

5. What is the influence of 10 % fetal bovine serum (that contains steroids) on the results since isothiocyanates can interact with steroid receptors?  What are the reasons this topic is not covered in this study? Especially in light of examining inflammation.

6.Section 2.9 Statistical Analysis- It is not appropriate to use a Students-t test when more than 2 treatments are being analyzed. Please re-analyze the data using an Analysis of Variance followed by pair-wise comparisons, where appropriate.

7. The discussion is a repeat of the results and only 4 references are cited.  Please re-write this section and discuss the present results in light of previous data reported by other investigators on 7MSI and related compounds and compare and contrast the present results with previous findings.

8. The true significance of this study is unknown based upon the poor presentation of the topic and especially the discussion section that is lacking.

Items 1, 3 4, 6 & 7 represent major issues in this evaluation for improvement

Author Response

The manuscript by Cho and Park on 7MSI for anti-inflammatory and autophagy effects on skin represents an exploratory in vitro investigation.  Several points should be addressed.

Comments 1: Title, this is an in vitro study, please change the title to: Anti-inflammatory and autophagy activation effects of 7-methylsulfonylheptyl isothiocyanate could suppress skin aging: in vitro evidence.

Response 1: Thank you for your insightful request. We have revised the title, "Anti-inflammatory and autophagy activation effects of 7-methylsulfonylheptyl isothiocyanate could suppress skin aging: in vitro evidence.”

Comments 2: Abstract – define NLRP3

Response 2: We have clarified the role and significance of NLRP3 (NOD-like receptor family, pyrin domain containing 3) in the Abstract. NLRP3 is a crucial inflammasome involved in regulating inflammatory responses, and our study addresses its activation and associated physiological effects.

Comments 3: The description of 7-MSI is not clear, please generate a figure with the chemical structure of this compound.

  1. References 21 and 22 about isothiocyanates is lacking, please include Navarro et al, Food & Function, 2011 as a reference
  2. There is no description of the abundance 7-MSI in cruciferous vegetables
  3. If 7-MSI can penetrate the skin layers
  4. How stable 7-MSI is or what is its solubility, etc.

Response 3: In the revised manuscript, we have addressed several key points based on the received feedback. We have introduced a new figure, now designated as Figure 1, which clearly illustrates the chemical structure of 7-MSI, enhancing the clarity of our descriptions. Additionally, we have incorporated the recommended reference from Navarro et al., published in Food & Function in 2011[23]. Regarding the abundance of 7-MSI in cruciferous vegetables, its levels vary, and specific data on its quantities in different cruciferous plants are limited. We have discussed that 7-methylsulfonylheptyl isothiocyanate (7-MSI) potentially can penetrate skin layers. However, effective penetration depends on its molecular properties, including size, polarity, and solubility, as well as the formulation used for delivery.

Comments 4: How do the authors justify the high concentrations of 7-MSI used in this in vitro study (micrograms/ml), while on the other hand propose that this compound may be a potential active ingredient for cosmetics?  Please include in the discussion the limitations of this in vitro study.

Response 4: We have added the MTS assay data as Supplementary Figure 1 in response to Reviewer 1's comments and your feedback.

Comments 5: What is the influence of 10 % fetal bovine serum (that contains steroids) on the results since isothiocyanates can interact with steroid receptors?  What are the reasons this topic is not covered in this study? Especially in light of examining inflammation.

Response 5: We appreciate the comment regarding the potential influence of 10% fetal bovine serum (FBS) containing steroids on our study, particularly concerning isothiocyanates interacting with steroid receptors. While FBS contains glucocorticoids and estrogens that can interact with these receptors, the concentration used in our study (10%) is standard for maintaining cell growth without significantly activating steroid receptors. Although residual steroid activity could be a minor confounding factor, we believe it is unlikely to substantially affect the specific outcomes from the isothiocyanate treatments.

Comments 6: Section 2.9 Statistical Analysis- It is not appropriate to use a Students-t test when more than 2 treatments are being analyzed. Please re-analyze the data using an Analysis of Variance followed by pair-wise comparisons, where appropriate.

Response 6: We agree with your opinion that using ANOVA is more appropriate when comparing multiple treatments rather than using a Student's t-test. However, in our study, the statistical analysis focused on comparing two groups: the group treated with 7-methylsulfonylheptyl isothiocyanate (7-MSI) and the untreated group, to analyze the effects of 7-MSI. Therefore, we employed a t-test for this specific comparison. We appreciate your understanding.

Comments 7: The discussion is a repeat of the results and only 4 references are cited.  Please re-write this section and discuss the present results in light of previous data reported by other investigators on 7MSI and related compounds and compare and contrast the present results with previous findings.

Response 7: Thank you for your constructive feedback on the discussion section of our manuscript. We recognize the need to provide a more comprehensive analysis that contextualizes our findings within the broader landscape of research on 7-MSI and related compounds.

To address your concerns, we have revised the discussion to incorporate a deeper analysis of our results in relation to previously published studies. Revised sections are marked in red.

Comments 8: The true significance of this study is unknown based upon the poor presentation of the topic and especially the discussion section that is lacking.

Response 8: To address this issue, we have thoroughly revised the discussion section to better articulate the implications and significance of our study. Revised sections are marked in red.

Round 2

Reviewer 1 Report

Thank you for improving the manuscript. Most of my comments could be addressed.

Line 195: The Supplementary Figure should not be shown in the main file but in the Supplement. This is only a matter of formatting.

Reviewer 2 Report

Authors have addressed review items

NA